# Particle Formation in a Complex Environment

**Doreena Dominick [1],\*, Stephen R. Wilson [1] , Clare Paton-Walsh [1] , Ruhi Humphries [1] , Élise-Andrée Guérette [1] , Melita Keywood [2] , Paul Selleck [2], Dagmar Kubistin [3] and Ben Marwick [4]**

[1] Centre for Atmospheric Chemistry, University of Wollongong, Wollongong, NSW 2522, Australia; swilson@uow.edu.au (S.R.W.); clarem@uow.edu.au (C.P.-W.); Ruhi.Humphries@csiro.au (R.H.); eag873@uowmail.edu.au (É.-A.G.)

[2] CSIRO Climate Science Centre, Aspendale 3195, Australia; Melita.Keywood@csiro.au (M.K.); Paul.W.Selleck@csiro.au (P.S.)

[3] German Meteorological Service, Meteorological Observatory Hohenpeissenberg, 82383 Hohenpeissenberg, Germany; kubid@gmx.de

[4] Center for Archaeological Science, University of Wollongong, Wollongong 2522, Australia; bmarwick@uw.edu

\* Correspondence: dd824@uowmail.edu.au; Tel.: +61-434-496-079

**Abstract:** A field aerosol measurement campaign as part of the Measurements of Urban, Marine and Biogenic Air (MUMBA) campaign was conducted between 16 January 2013 and 15 February 2013 in the coastal city of Wollongong, Australia. The objectives of this research were to study the occurrence frequency, characteristics and factors that influence new particle formation processes. Particle formation and growth events were observed from particle number size distribution data in the range of 14 nm–660 nm measured using a scanning particle mobility sizer (SMPS). Four weak Class I particle formation and growth event days were observed, which is equivalent to 13% of the total observation days. The events occurred during the day, starting after 8:30 Australian Eastern Standard time with an average duration of five hours. The events also appeared to be positively linked to the prevailing easterly to north easterly sea breezes that carry pollutants from sources in and around Sydney. This suggests that photochemical reactions and a combination of oceanic and anthropogenic air masses are among the factors that influenced these events.

**Keywords:** new particle formation; Southern Hemisphere; Australia; atmospheric aerosol; MUMBA

## 1. Introduction

Atmospheric aerosols have a major influence on health and climate [1–3]. Recently, Cohen et al. [4] reported that ambient fine aerosols ($PM_{2.5}$) were the fifth ranking mortality risk factor in 2015, where the global mortality estimated due to ambient $PM_{2.5}$ was 4.2 million people annually. Studies have identified a correlation between fine particulate matter and various health conditions, particularly respiratory and cardiovascular disease [5,6]. The International Agency for Research on Cancer (IARC) had defined particulate matter as a human carcinogen [7]. In addition, the Earth's climate is influenced both directly and indirectly by atmospheric particles. Particles scatter and absorb radiation [3,8,9]. Indirectly, they act as cloud condensation nuclei and influence the characteristics of clouds [10].

New particle formation is an important source of atmospheric aerosols and plays a key role in influencing the properties of aerosol particles [11]. Atmospheric new particle formation (NPF) events involve the formation of stable particle clusters (new particles of approximately 1 nm) [12], followed by growth to a detectable size. The detectable particle size ranges from 1 nm–10 nm [12,13]. Both natural and anthropogenic sources contribute to new particle formation processes [14] through the formation

of chemical species such as photochemical oxidation products, for example $H_2SO_4$ [15], oxidation products of volatile organic compounds [16–18], $NH_3$ [19], and amines [20].

Geographical and topographical factors influence the transport and build up of species that may be related to particle formation and growth events [21]. Kulmala et al. [13] reported that many studies have been conducted globally in order to investigate particle formation processes in different environmental settings. However, most of the studies have been conducted in the Northern Hemisphere, at various locations from clean, remote sites to polluted environments (e.g., [12,22–26]). There were also global model studies of new particle formation, for example Yu and Luo [27], Pierce and Adams [28], Spracklen et al. [29], Gordon et al. [30]. The Northern Hemisphere contains the majority of the Earth's land mass (approximately 68%), while the Southern Hemisphere mainly consists of oceans. This means that the Northern Hemisphere is more heavily populated (approximately 90% of the global population) and is more industrialised. There are also differences in vegetation between the Northern Hemisphere and the Southern Hemisphere [31]. These differences are likely to result in different aerosol properties in the Southern Hemisphere. Hence, more studies are required to understand the characteristics of aerosols in the Southern Hemisphere [32,33]. A recent study on NPF events in the urban environment in northeastern Australia by Pushpawela et al. [34] reported that 41% of the days contained NPF events. These were identified in one year of measurements using a neutral cluster and air ion spectrometer (NAIS). The highest occurrence of NPF events was during summer, with a starting time between 8:00 and 8:30. Other previous NPF studies in the urban environment in Australia include Salimi et al. [35], Cheung et al. [36] and Mejia et al. [37]. Suni et al. [38] conducted a study on the particle formation associated with natural emissions from the eucalyptus forest in southeast Australia. Modini et al. [39] focussed on the particle size distribution in a sub-tropical clean marine site, and Cainey et al. [40] studied the characteristics of particle size at a clean marine mid-latitude site. Guo et al. [41] conducted a study on the size distribution and particle formation in the rural environment of eastern Australia.

To further investigate the characteristics of atmospheric aerosols in the Southern Hemisphere, particularly Australia, the study presented here uses data from a campaign known as Measurements of Urban, Marine and Biogenic Air (MUMBA). This campaign focused on the air quality in a coastal urban environment influenced by mixed emissions from the ocean, the forests and urban environments. A previous study on the characteristics of particle number during the MUMBA campaign by Dominick et al. [42] noted that the population of the particle number was dominated by ultra-fine particles (particles with diameters ranging from 3 nm–100 nm). Ultra-fine particles contribute to a large fraction of the total particle number concentration globally [13,43]. New particle formation is one of the sources of ultra-fine particles [44]. Therefore, the objective of this research was to study the occurrence frequency, characteristics and factors that promote the new particle formation processes during the MUMBA campaign. Details of this campaign can be found elsewhere [42,45–47].

## 2. Materials and Methods

### 2.1. Site Description and Weather Conditions

The MUMBA campaign was carried out at 34.397° S and 150.900° E, within the University of Wollongong, Campus East (Figure 1), in the Illawarra region of New South Wales, eastern Australia. This campaign ran for approximately two months (21 December 2012–15 February 2013). However, the aerosol measurements were made for one month (16 January 2013 and 15 February 2013). Wollongong is New South Wales' third largest city after Sydney and Newcastle. It is also the tenth largest city in Australia.

Weather conditions experienced during the aerosol measurement period were dominated by clear days. Temperature ranged from 14 °C–44 °C, and relative humidity ranged from 19–96%. The general pattern of the wind direction observed during the campaign was westerly winds overnight and early

in the morning and easterly winds in the middle of the day. Low wind speeds generally occurred in the morning, increasing in the middle of the day and decreasing for the rest of the day [42,45,46].

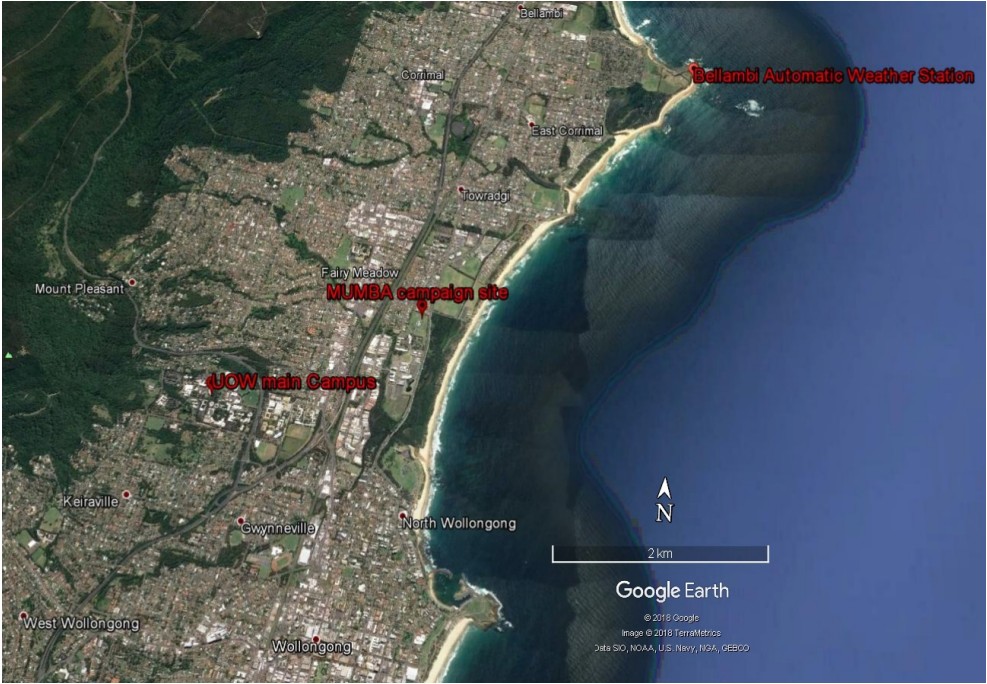

**Figure 1.** Map of the Measurements of Urban, Marine and Biogenic Air (MUMBA) campaign site, Bellambi Automatic Weather station and University of Wollongong (UOW) main campus. The west and east directions of the MUMBA campaign site were forest and ocean environments, respectively. The north and south directions of the campaign site were urban environments Generated in Google Earth®.

### 2.2. Instrumentation and Data Analysis

The measurement of the particle number size distribution of the aerosol particles was performed with a scanning mobility particle sizer (SMPS) spectrometer (TSI 3936). This instrument measures the number of particles within 64 size bins with diameters from 14 nm–660 nm with a measurement time resolution of approximately five minutes. The three major components of the SMPS spectrometer are: (i) an electrostatic classifier (TSI 3080); (ii) a differential mobility analyser (TSI 3081); and (iii) a condensation particle counter (TSI 3772). Butanol was used as the condensing vapour in the condensation particle counter. In addition, total concentrations of particles between 3 nm and 2.5 $\mu$m ($CN_3$) and cloud condensation nuclei (CCN) were also measured. $CN_3$ measurements were made using an ultra-fine condensation particle counter (TSI Model 3776, Shoreview, MN, USA), while CCN measurements were made using the Droplet Measurement Technologies (Longmont, CO, USA) operated at 0.55% supersaturation. Air was sampled through an 8 m-long copper inlet for both $CN_3$ and CCN measurements. The cut-off diameters and calibration techniques were performed as stated by the manufacturers. Auxiliary data included $NO_x$, CO, $O_3$, $SO_4^{2-}$, and volatile organic compounds (such as isoprene, benzene, toluene, xylenes) and meteorological data including temperature, global irradiance, wind speed, wind direction and relative humidity. Paton-Walsh et al. [45] and Guérette et al. [46] have described in detail the instrumentation, as well as the data processing. Data obtained from this campaign have been published online at PANGAEA (https://doi.pangaea.de/10.1594/PANGAEA.871982). It should be noted that global irradiance was measured using a solar radiation sensor (Davis Instruments, Vintage Pro2, Hayward, CA, USA) located at the University of Wollongong (Figure 1).

The origin of air masses arriving at the monitoring site was studied using the hybrid single-particle Lagrangian integrated trajectory (HYSPLIT) model [48,49]. The Global Data Assimilation System

(GDAS) analysis product, a ready formatted meteorological dataset, was used as the model data input. This dataset is publicly available from the National Oceanic and Atmospheric Administration (NOAA) at ftp://ftpprd.ncep.noaa.gov//pub/data/nccf/com/hysplit/prod/, the National Centre for Environmental Prediction (NCEP) [50] and at www.ready.noaa.gov/archives.php for NOAA Air Resources Laboratories (NOAA ARL). Back-trajectory calculations are one of the most widely-used features in HYSPLIT [51].

Descriptive and statistical analyses were carried out using R statistical analysis Version 3.5.2 [52]. The main R package used in the statistical analysis was "openair" Version 2.4.2 [53].

*2.3. Classification, Growth Rate and Condensation Sinks of Particle Formation Events*

Atmospheric new particle formation and growth involve a burst of particles growing from small into larger size particles by the process of condensation and or coagulation [13]. The particle formation events observed in this work focused on Class I events [54]. The criteria for Class I used here were: (i) a new mode of particles (30 nm) must be observed; (ii) the event must occur over a time span of hours; (iii) the new mode must show signs of growth. When plotted as a contour plot of particle number as a function of time and particle diameter, a Class I event must show a clear particle formation pattern called a "banana shape" [54]. The classification in this work here was performed visually.

The strength of Class I particle formation and growth event days can be further classified using the method used by Zhang et al. [55]. The classification used the rate of change of total particle number concentration ($dN/dt$), where $N$ is the number of particles in the nucleation mode (i.e., particles with a diameter less than 10 nm). Particles in the size range of 3 nm–14 nm were used in this study, and events were classified into two classes:

(a)　A "strong" particle formation event is when:
　　　$N > 10{,}000 \text{ cm}^{-3}$ for at least 1 h and
　　　$dN/dt > 10{,}000 \text{ cm}^{-3} \text{ h}^{-1}$
(b)　A "weak" particle formation event is when:
　　　$5000 < N < 10{,}000 \text{ cm}^{-3}$ for at least 1 h and
　　　$5000 < dN/dt < 10{,}000 \text{ cm}^{-3} \text{ h}^{-1}$

Growth rate is one of the most relevant variables in identifying particle formation events, and it is defined as the change in particle diameter due to particle population growth [56]. This growth rate was computed based on the appearance time method [57] (Equation (1)). The geometric mean diameter (GMD) was used to define a "characteristic size". GMD values were retrieved using the Aerosol Instrument Management software, Version 10.2.0.11 by TSI, Shoreview, MN, USA.

$$\text{Growth rate} = \frac{(D_{pg2} - D_{pg1})}{(t_2 - t_1)} \tag{1}$$

Here, $D_{pg2}$ and $D_{pg1}$ represent the geometric mean diameter at time $t_2$ and $t_1$, respectively, when a clear particle formation pattern is observed.

The condensation sink (CS) is defined as the loss rate of condensable vapour molecules condensed on pre-existing aerosols [58]. The CS value can be calculated as [59]:

$$\text{CS} = 2\pi D \sum_i \beta_m(d_{p,i}) d_{p,i} N_i \tag{2}$$

where $D$ is the diffusion coefficient of the condensing vapour and $\beta_m$ is the mass flux transition correction factor. $d_i$ and $N_i$ respectively represent the particle diameter and particle number concentration in the size bin.

Sulphuric acid is the primary condensable vapour species, and the diffusion coefficient of sulphuric acid can be calculated using Equation (3) [56,59,60]:

$$D = 5.0032 \times 10^{-6} + 1.04 \times 10^{-8} T + 1.64 \times 10^{-11} T^2 - 1.566 \times 10^{-14} T^3 \tag{3}$$

Here, $T$ is the temperature in Kelvin. In this study, we used $T = 297.5$ K, which was the average maximum temperature in Wollongong during the summer of 2012/2013.

The $\beta_m$ is typically calculated using the expression given by Fuchs and Sutugin [61]:

$$\beta = \frac{1 + K_n}{1 + 1.677 K_n + 1.333 K_n{}^2} \tag{4}$$

where $K_n$ is the Knudsen number, which can be calculated using:

$$K_n = \frac{2\lambda}{d_p} \tag{5}$$

Here, $\lambda$ is the effective mean free path of the atmosphere. In this study, $\lambda = 108$ nm was used, as also used by Kulmala et al. [57] and Massman [62].

## 3. Results and Discussion

### 3.1. General Characteristics of Class I Particle Formation Events

Particle formation events were identified visually during the 31 sampling days of this campaign. There were four Class I event days identified, with one event in January 2013 (22 January) and three more observed in February 2013 (6, 7 and 8 February) (Table 1 and Figure 2a). The occurrence frequency of particle formation events was therefore 13% of the total number of days. All Class I events occurred during the daytime, approximately between 8:30 and 14:30, with an average time duration of 5 h, shown in Figure 2a. All the identified Class I event days were classified as "weak" particle formation events. During the Class I event days, total particle number concentration increased from approximately 3000 cm$^{-3}$–15,000 cm$^{-3}$.

**Table 1.** Summary of the Class I particle formation event days during the MUMBA aerosol measurements period. GMD is the geometric mean diameter, and GR is growth rate as defined in Equation (1). The criteria for classification are discussed in Section 2.3.

| Date | Time (Start) | Time (End) | Duration (hours) | Primary GMD (nm) | Final GMD (nm) | GR (nm/h) | Classification |
|------|------|------|------|------|------|------|------|
| 22 January 2013 | 8:30 | 14:00 | 6.3 | 30.0 | 70.0 | 6.3 | Weak |
| 6 February 2013 | 10:00 | 13:00 | 3.0 | 25.0 | 50.0 | 8.3 | Weak |
| 7 February 2013 | 10:00 | 14:00 | 4.0 | 24.0 | 55.0 | 7.8 | Weak |
| 8 February 2013 | 8:30 | 14:30 | 6.0 | 30.0 | 60.0 | 5.0 | Weak |
| Average | | | 4.8 | 27.2 | 58.8 | 6.9 | |

Particle growth rates in this study were between 5.0 nm h$^{-1}$ and 8.3 nm h$^{-1}$ with an average value of 6.9 nm h$^{-1}$. These observed growth rates fall in the range reported for other urban environments (0.5 nm h$^{-1}$–9.0 nm h$^{-1}$ [13]) and are also comparable to the values reported from a remote, sub-tropical coastal environment in Australia (1.8 nm h$^{-1}$–8.2 nm h$^{-1}$ with an average value of 6.4 nm h$^{-1}$) [39].

For a more complete comparison, the Class II new particle formation events measured during MUMBA were also identified. A Class II event is where particle growth is observed, but due to fluctuations in diameter with time, the growth and formation rate is difficult to quantify uniquely [54]. There were four Class II new particle formation events identified (17, 27, 29 January and 9 February 2013) (Figure 2b), which gave a total of eight particle formation events identified in this period. These eight days therefore give the occurrence of 25% over the total number of sampling days. For comparison, new particle formation events measured in Sydney during summer occurred on 50% of the sampling days during the Sydney Particle Study (Class I and Class II) Cope et al. [63].

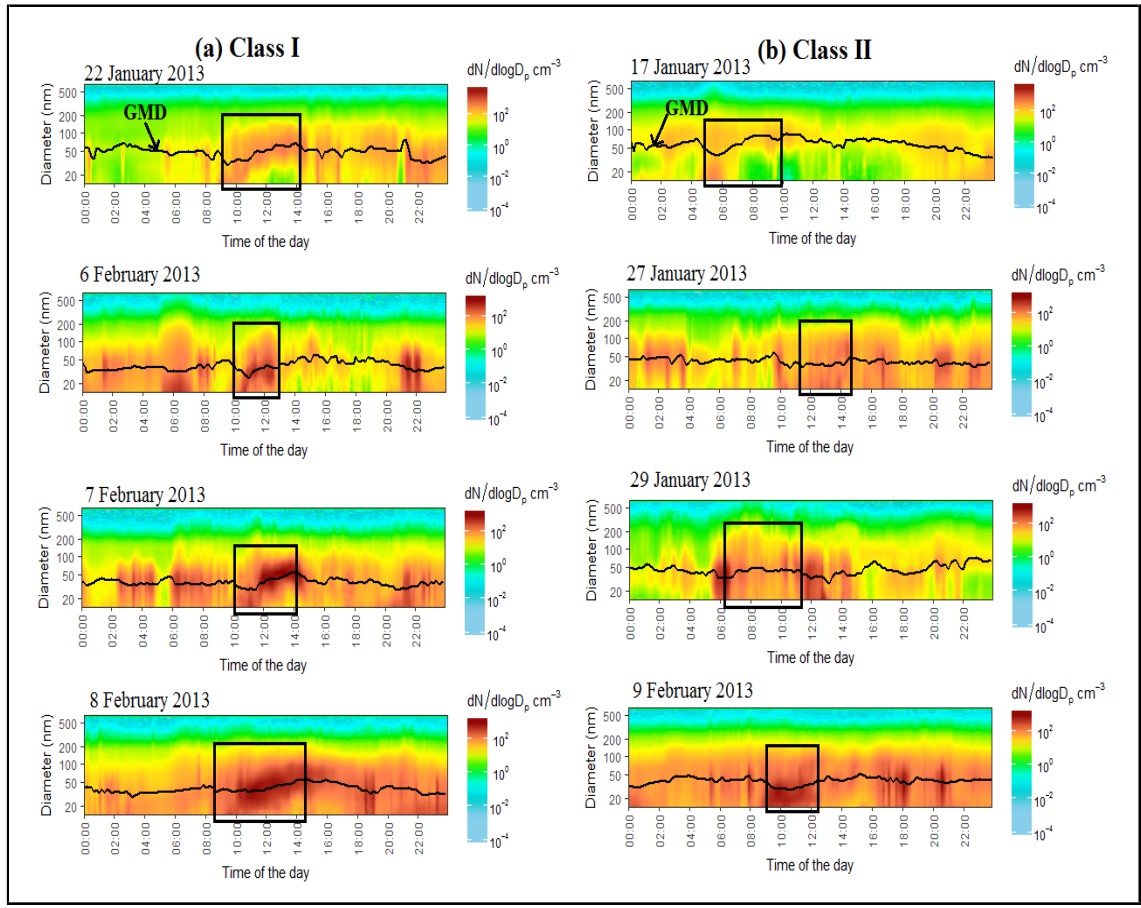

**Figure 2.** Contour plots of particle number as a function of time and particle diameter for the (**a**) Class I and (**b**) Class II event days identified during the MUMBA campaign. The black line is the geometric mean diameter (GMD). The black rectangular box represents the relevant time of particle formation events. Ten-minute averaged data were used. Note that the y-axis scale is logarithmic.

The 24-h backward trajectories plotted for the entire aerosol measurement period revealed that 50% of the sampled air masses arriving at 10:00 Australian Eastern Standard Time, (AEST) were from the north and northeast sectors. 10:00 AEST was chosen as the arriving time for the trajectory because the particle formation and growth events (Class I and Class II) typically began around this time. The Class I and Class II events always occurred on the days when the wind was from the north and northeast sectors. This observation indicates that air mass from the north and northeast sectors is important for the occurrence of the Class I and Class II event days identified during the MUMBA campaign.

The average CS calculated prior to the Class I and Class II events is shown in Table 2. Overall, Class II event days recorded higher CS, where the highest CS occurred on 9 February 2013.

**Table 2.** Condensation sink during the Class I and Class II events during the MUMBA campaign.

| Class I | Condensation Sink, CS [a] $(s^{-1})$ $(10^{-2})$ | Class II | Condensation Sink (CS) [a] $(s^{-1})$ $(10^{-2})$ |
|---|---|---|---|
| 22 January 2013 | $0.6 \pm 0.3$ | 17 January 2013 | $0.9 \pm 0.2$ |
| 6 February 2013 | $0.2 \pm 0.03$ | 27 January 2013 | $0.5 \pm 0.2$ |
| 7 February 2013 | $0.3 \pm 0.1$ | 29 January 2013 | $0.6 \pm 0.3$ |
| 8 February 2013 | $0.6 \pm 0.2$ | 9 February 2013 | $1.0 \pm 0.2$ |

[a] The condensation sink was averaged 1 h prior to the start of the particle formation and growth event.

### 3.2. Factors that Influence Particle Formation and Growth Events

#### 3.2.1. Sulfate ($SO_4^{2-}$)

Measurements of total $SO_4^{2-}$ were made from 22 January 2013–14 February 2013 during the MUMBA campaign using a high volume sampler (details in Paton-Walsh et al. [45]). The high volume sampling times were morning (AM) (4:00–9:00) and afternoon (PM) (10:00–18:00). Note that by this time definition, the particle formation and growth occurred in the PM period. The concentration of total $SO_4^{2-}$ in the afternoon (PM) was higher during the Class I event days compared to the non-event days (Figure 3). This could indicate that $SO_2$ played a role in particle formation and particle growth at this monitoring site.

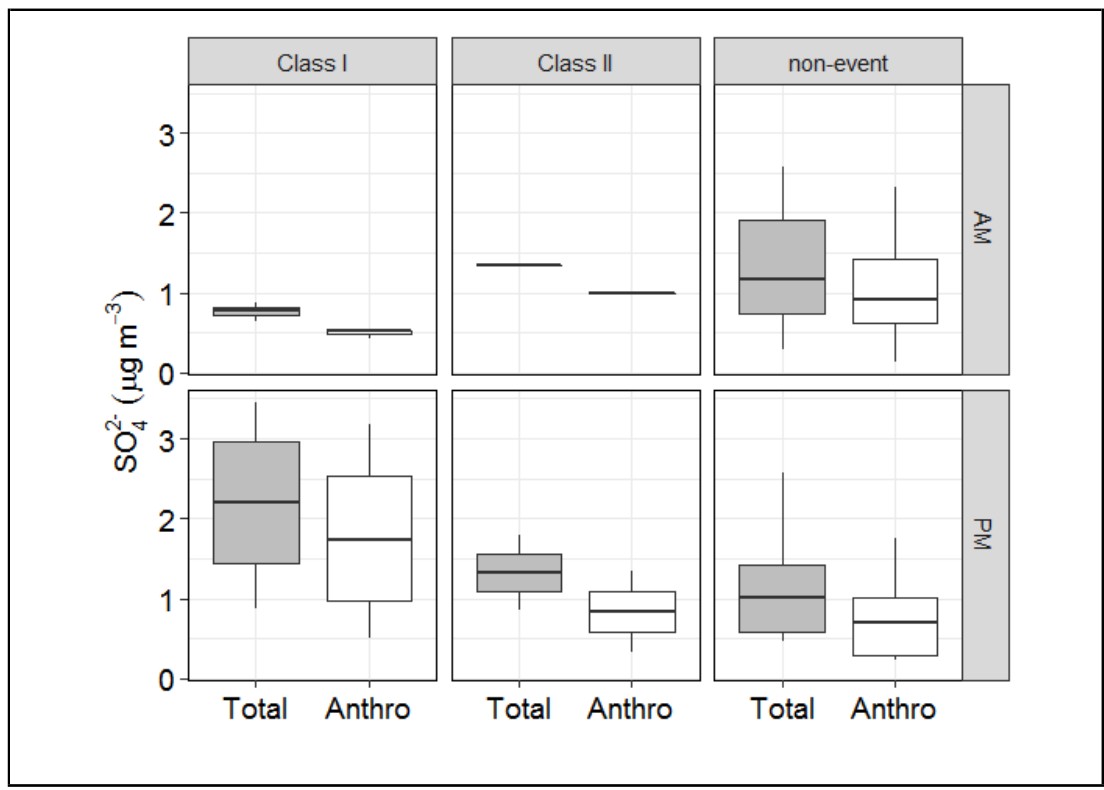

**Figure 3.** Box plot of total ("Total") and anthropogenically- ("Anthro") derived $SO_4^{2-}$ on Class I event days (22 January, 6 February, 7 February and 8 February 2013, number of samples = 14), Class II event days (29 January and 9 February 2013, number of samples = 6) and on non-event days (number of samples = 64). "AM" and "PM" refer to measurements from 4:00–9:00 and 10:00–18:00, respectively. The central line of each box is the median. The lower and upper edges of each box are the 25% quartile and 75% quartile, respectively. Whiskers are extended to the smallest of either the furthest data point or 1.5-times the quartile value.

Condensation sink rates during the period when $SO_4^{2-}$ measurements were made were calculated and are illustrated in Table 3. The highest CS occurred during the PM period of Class I event days, with a median value and standard deviation of 1.18 ($\pm$ 0.06) $\times 10^{-2}$ s$^{-1}$. Kulmala et al. [64] has reported that higher CS rates can suppress particle formation and growth. However, high CS rates during the Class I events were not a limiting factor for the occurrence of particle formation and growth. This is presumably because the gaseous precursors for particle formation partially cancel out the effect of the larger CS [65].

**Table 3.** Summary of the condensation sink (CS) on the Class I event days, the Class II event days and non-event days during the $SO_4^{2-}$ measurement period.

| | | Condensation Sink (s$^{-1}$) (10$^{-2}$) | | | | |
|---|---|---|---|---|---|---|
| | | Minimum | Maximum | Mean | Median | Std |
| Class I [b] | AM [a] | 0.11 | 1.49 | 0.59 | 0.54 | 0.03 |
| | PM [a] | 0.10 | 3.24 | 1.35 | 1.18 | 0.06 |
| Class II [b] | AM [a] | 0.09 | 2.62 | 0.89 | 0.83 | 0.06 |
| | PM [a] | 0.18 | 3.17 | 0.90 | 0.84 | 0.06 |
| Non_event [b] | AM [a] | 0.11 | 3.47 | 0.73 | 0.58 | 0.02 |
| | PM [a] | 0.13 | 0.13 | 0.35 | 0.22 | 0.03 |

[a] Condensation sink for "AM" and "PM" refers to the average of CS from 4:00–9:00 and 10:00–18:00, respectively. [b] Class I event days (22 January, 6 February, 7 February and 8 February 2013), Class II event days (29 January and 9 February 2013 and on non-event days.

Sea salt aerosols were expected to be observed during the MUMBA campaign since the measurement site is located close to the open ocean. Sea salt aerosols are made up of around 33% sodium chloride (NaCl) [66]. In addition to NaCl, other chemical ions in sea salt include $SO_4^{2-}$, $Ca^{2+}$, $Mg^{2+}$ and methane sulfonic acid (MSA). To study the sources of $SO_4^{2-}$, the method used by Millero et al. [67] for seawater composition was applied. Sulfate makes up 2.7 g of the 35.2 g salt content in 1000 g of seawater (nearly 8% of sea salt). As the magnesium ion ($Mg^{2+}$) concentration in seawater is a conservative major element, $Mg^{2+}$ can be used to calculate $SO_4^{2-}$(ss) (Equation (6)), where "ss" is sea salt sulfate.

$$SO_4^{2-}(ss) = Mg^{2+} \times \frac{2.7}{1.2} \tag{6}$$

The anthropogenically-derived $SO_4^{2-}$ can be calculated by assuming that there are three dominant sources: anthropogenic, sea salt and biogenic from the ocean, as illustrated in Equation (7), where, "Anthro" is anthropogenic sulfate and "BiogenicOceanic" is sulfate from phytoplankton:

$$SO_4^{2-}(total) = SO_4^{2-}(Anthro) + SO_4^{2-}(ss) + SO_4^{2-}(BiogenicOceanic)$$
$$SO_4^{2-}(Anthro) = SO_4^{2-}(total) - SO_4^{2-}(ss) - SO_4^{2-}(BiogenicOcenic) \tag{7}$$

"BiogenicOceanic" can be estimated using Equation (8) by assuming a $SO_4^{2-}$ contribution at a fixed ratio to MSA [68].

$$SO_4^{2-}(BiogenicOceanic) = MSA \times 5.1 \tag{8}$$

The MSA is derived from the oxidation of dimethyl sulphide (DMS) emitted from phytoplankton [68]. The value of 5.1 g ($SO_4^{2-}$(ss))/(g MSA) was the average ratio measured in $PM_{10}$ at Cape Grim in February over the past 10 years (2002–2012) [69]. February was selected to represent summer in Australia. The Cape Grim Baseline Air Pollution station is exposed to Southern Ocean clean air [70]. At Cape Grim during clean air conditions, the primary known source of $SO_4^{2-}$(ss) and MSA is from the oxidation of DMS. Therefore, the ratio of MSA to $SO_4^{2-}$(ss) from Cape Grim can be used to infer the amount of $SO_4^{2-}$(ss) from DMS oxidation (sulfate from phytoplankton) during the MUMBA campaign, as illustrated in Equation (8). Using Equation (7), we estimated that 70% of the sulfate was from anthropogenic sources, 20% from sea salt and 10% from biogenic sources. The composition of sea salt is well established, and even a 50% error in the biogenic contribution would only change the anthropogenic estimate by 5%. We therefore ascribe a conservative estimate of the uncertainty in the anthropogenic fraction of 10%. The fact that the concentration of anthropogenically-derived $SO_4^{2-}$ was higher in the afternoon of the Class I event days compared to the non-event days suggests that anthropogenic $SO_4^{2-}$ is important for the new particle formation and growth (Figure 3).

### 3.2.2. Particle Number Size Distributions and Condensation Sinks

This section examines the characteristics of the particle number observed between event days of Class I and Class II events. Here, we focus on 9 January 2013 as a day of a Class II event because it was preceded by three days containing Class I events (6 February–8 February 2013). These four days (6 February–9 February 2013) experienced similar meteorological conditions particularly in terms of the sources of air masses as identified by back trajectories (Figure 6) and global irradiance (Figure 5). Only measurements from 7:00–15:00 AEST were used in this analysis.

To investigate if there was a difference in size distributions between Class I event days and the selected Class II event day, a particle number ranging from 3 nm–100 nm was considered. As illustrated in the contour plots of the Class I event days, particle formation was observed in a particle size range less than 100 nm (Figure 2). Particle number concentrations for diameters ranging from 3 nm–14 nm ($PNC_{3nm–14nm}$) were used to represent smaller size particles. Meanwhile, particles ranging from 14 nm to 100 nm ($PNC_{14nm–100nm}$) were used to represent the larger size particles.

Figure 4a,b shows the comparison of $PNC_{3nm–14nm}$ and $PNC_{14nm–100nm}$ for the Class I event days and the selected Class II event day. Both the Class I event days and the selected Class II event day recorded $PNC_{3nm–14nm}$ below $10 \times 10^3$ cm$^{-3}$, except for 6 February 2013, which reached the maximum particle number ($50 \times 10^3$ cm$^{-3}$) at 11:00 (Figure 4a). The $PNC_{14nm–100nm}$ steadily increased from 9:00 for both Class I event days and the selected Class II event day. The $PNC_{14nm–100nm}$ on the Class I event days began to decrease at around 13:00, and there was an early decrease (after 10:00) in the particle number of $PNC_{14nm–100nm}$ observed on the selected Class II event day (Figure 4b). There was a sharp peak observed between 8:00 and 9:00 for $PNC_{14nm–100nm}$ on the selected Class II event day with a particle number concentration ranging from $4 \times 10^3$ cm$^{-3}$–$20 \times 10^3$ cm$^{-3}$. The $PNC_{14nm–100nm}$ for Class I event days between 8:00 and 9:00 ranged from $1 \times 10^3$ cm$^{-3}$–$9 \times 10^3$ cm$^{-3}$ (Figure 4b).

On the selected Class II event day, there was an increase in the $PNC_{3nm–14nm}/PNC_{14nm–100nm}$ ratio just after 8:00, which indicated that the particle number of smaller particles ($3$ nm $< Dp < 14$ nm) was higher than that of larger particles ($PNC_{14nm–660nm}$) (Figure 4c). However, instead of a decrease in the $PNC_{3nm–14nm}/PNC_{14nm–100nm}$ ratio and an increase in $PNC_{14nm–660nm}$, as observed on Class I event days (at least on 6 and 8 February 2013, where observations were complete), both the larger and smaller size particles continued to increase in number (i.e., increase in both $PNC_{3nm–14nm}/PNC_{14nm–100nm}$ and $PNC_{14nm–660nm}$). This was followed by an unclear particle growth process in the contour plot from approximately 9:00 (Figure 1). Based on this observation, it seems that there was a high number of smaller particles; however, these particles did not grow into larger sized particles. In terms of CS, 9 February 2013 recorded the highest CS compared to the CS recorded on 6 February, 7 February and 8 February 2013 (Table 2). High CS is associated with high pre-existing particles. Pre-existing particles provide a significant sink for condensable vapours in the atmosphere, which may prevent particle formation and growth [71]. High CS rates can be one of the factors that drove the observed Class II particle formation and growth on 9 February 2013.

### 3.2.3. Meteorological Conditions

Local meteorological variables including temperature, relative humidity, wind speed, wind direction and global irradiation have been analysed to study their relationship to the particle formation and growth events. As some of the meteorological data measured on site during the MUMBA campaign site were missing at relevant times, meteorological variables measured by the Automatic Weather Station operated by the Australian Bureau of Meteorology at Bellambi (about 4 km northeast of the MUMBA campaign site) (Figure 1) were used. An analysis of the wind data from the MUMBA campaign site and from the Bellambi weather stations demonstrated that both stations experienced similar meteorological conditions [72]. The non-event days used in this work covered 22 days of the 31 sampling days and excluded the four Class I event days, the four Class II event days and the hottest day (18 January 2013) of the entire aerosol measurements period.

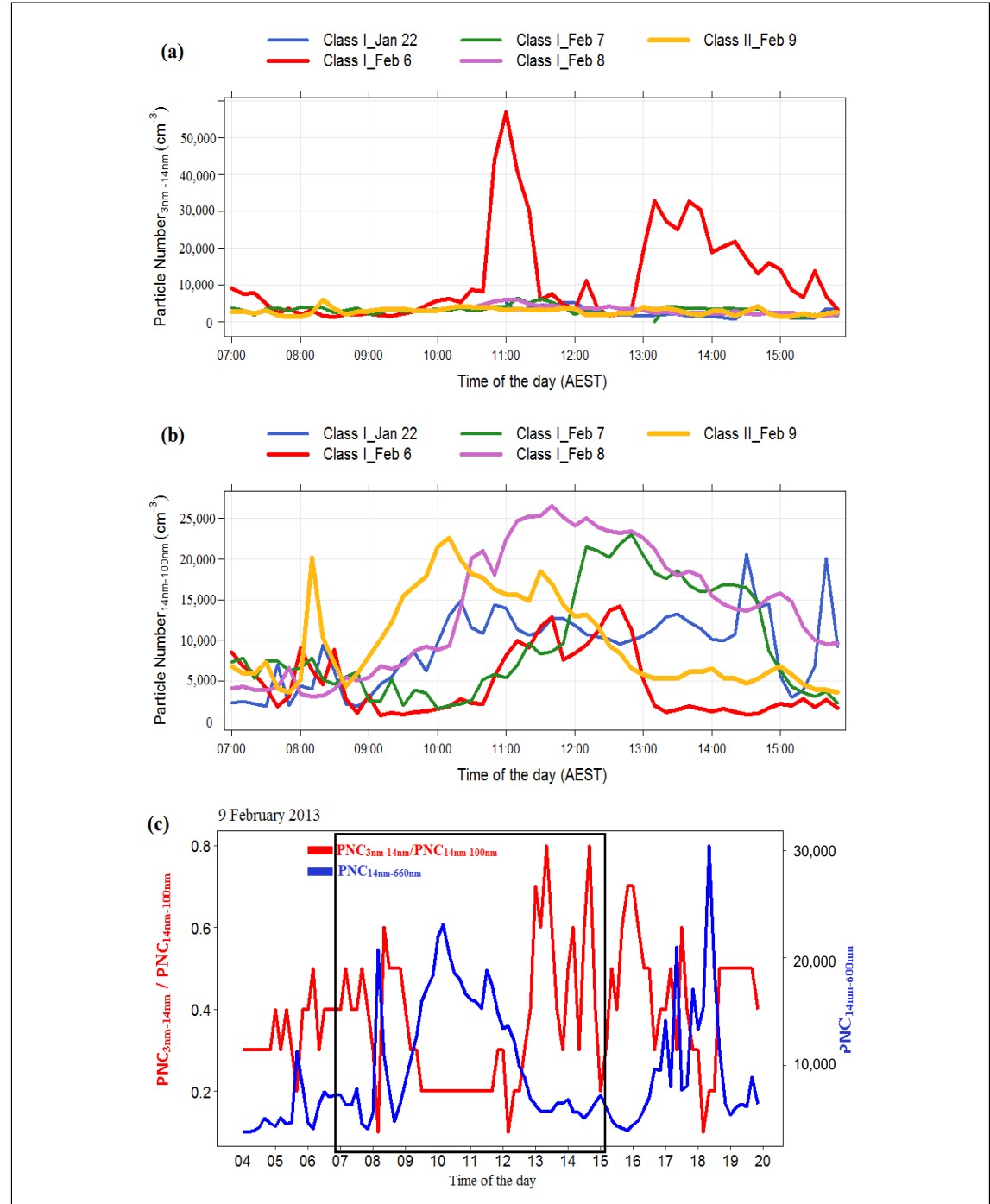

**Figure 4.** Time series of (**a**) Particle number concentrations for diameters ranging from 3 nm–14 nm ($PNC_{3nm–14nm}$) and (**b**) $PNC_{14nm–100nm}$ on the Class I event days (22 January, 6 February, 7 February and 8 February 2013) and on the selected Class II event (9 February 2013). (**c**) Time series of the total particle number between 14 nm and 660 nm ($PNC_{14nm–660nm}$) and the ratio of particle number ranging from 3 nm–14 nm and particle number ranging from 14 nm–100 nm (($PNC_{3nm–14nm}$)/$PNC_{14nm–100nm}$) on 9 February 2013. The black rectangular box represents the relevant time of particle populations before, during and after the Class II event. Ten-minute averaged data were used for (**a,b**). Five-minute averaged data were used in (**c**).

As illustrated in Figure 5, air temperature on the Class I event days ranged from 20 °C–25 °C with an average of 24 °C. On the non-event days, the average temperature was 22 °C. Relative humidity on

the Class I varied between 65% and 75% with an average value of 70%; meanwhile, non-event days recorded an average value of 77%. The average wind speed was 7 m s$^{-1}$ with a minimum of 4 m s$^{-1}$ and a maximum of 10 m s$^{-1}$. Generally, it was sunny during the event days, except for broken clouds in the morning of 6 February 2013 and in the afternoon of 22 January 2013. The maximum global irradiance recorded on the event days and non-event days was around 1000 W m$^{-2}$ and 600 W m$^{-2}$, respectively. Of the four event days, relative humidity was higher on 22 January and 8 February 2013 than on 6 February and 7 February 2013 (Figure 5). Local wind directions during Class I event days were dominated by air masses from the north and northeast directions (Figure 5).

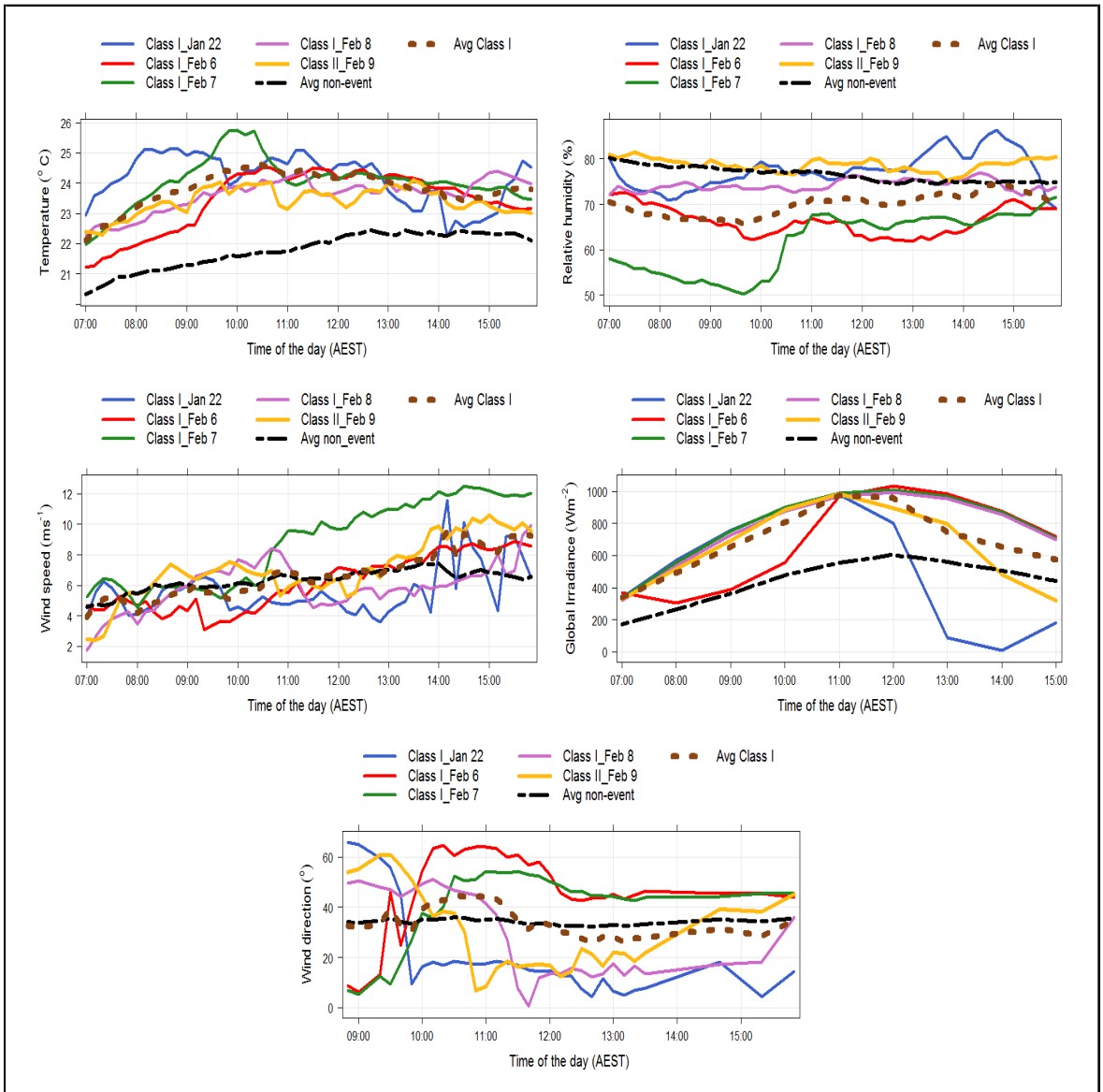

**Figure 5.** Time series of temperature, relative humidity, wind speed, wind direction and global irradiance on Class I event days (22 January, 6 February, 7 February and 8 February 2013), a selected Class II event day (9 February 2013). Average value of meteorological variables on non-event days and Class I event days are also included. Ten-minute averaged data were used in these plots except for global irradiation, which was averaged hourly. Only measurements from 7:00–15:00 AEST are plotted.

The time series of meteorological variables during the four Class I event days was compared with that of 9 February 2013 (selected Class II event) (Figure 5). The temperature and wind speed were very similar for all days. The global irradiance on the selected Class II event day was similar

to that observed on 22 January 2013. Local air masses experienced during the Class I event days and the selected Class II event day were from the north and northeast sectors (Figure 5). Within this north and northeast sectors, there were some changes in the local wind directions around 9:00–11:00 (Figure 5). This could be due to the temperature difference between the land and the ocean (sea breeze). The selected Class II event day experienced stable relative humidity between 7:30 and 9:00, at around 80%. Relative humidity experienced on Class I event days between 7:30 and 9:00 ranged from 55–75%.

The 24-h backward trajectories produced by HYSPLIT arriving at 10:00 AEST at 100 m above the site indicated that air masses (during the Class I) arrived at the sampling point after travelling over the South Pacific Ocean and populated urban areas including Sydney (Figure 6). Air masses from the ocean and urban areas (from north and northeast directions) could be one of the factors that influenced the occurrence of the observed Class I events.

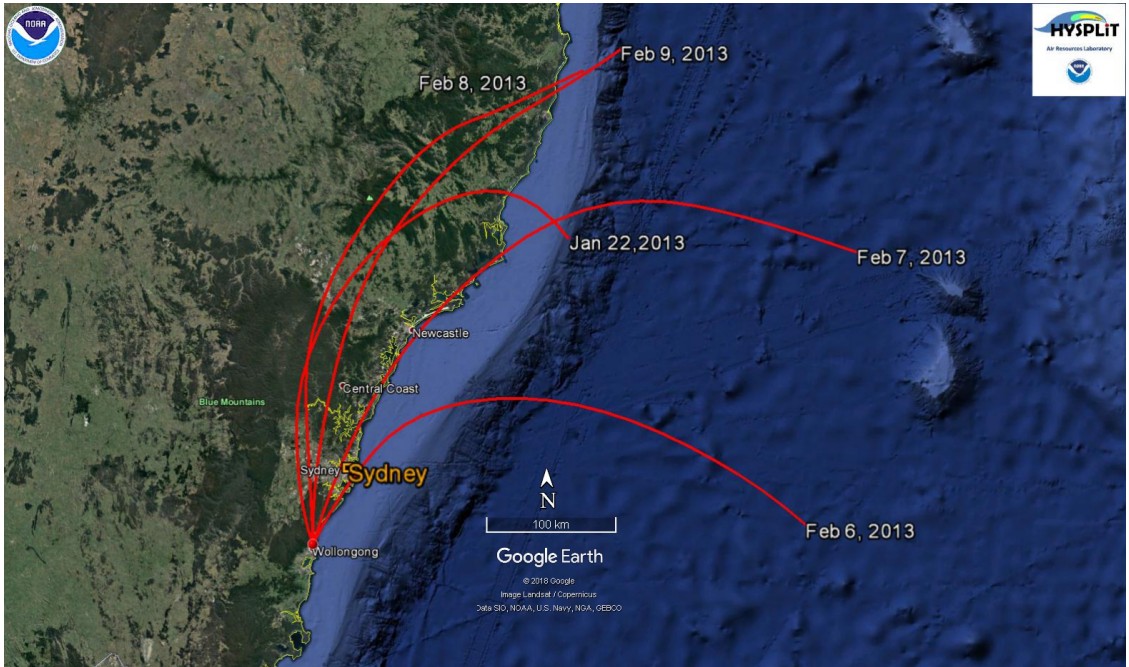

**Figure 6.** The 24-h backward trajectories produced by HYSPLIT at 100 m above the site for Class I event days (22 January, 6–8 February 2013) and the selected Class II event day (9 February 2013). The monitoring site received the air masses at 10:00 AEST (+10 UTC). Generated in Google Earth®.

The local meteorological variables therefore were similar during the Class I and the selected Class II event day measurement periods, except for a slightly higher relative humidity on the selected Class II event day. Relative humidity can interfere with particle growth processes [73]. However, the difference in relative humidity was not large enough to draw a conclusion with any confidence.

### 3.2.4. CO, $NO_x$, $O_3$ and Isoprene

The concentrations of combustion markers such as CO and $NO_x$ and the formation of secondary pollutants from photochemical reactions (marked by ozone ($O_3$)) at the time relevant for when the events occurred (7:00–15:00 AEST) are illustrated in Figure 7. Concentrations of CO and $NO_x$ increased before the particle formation and growth events. The concentrations observed for CO and $NO_x$ were lower on 6 February and 7 February 2013 between 8:00 and 10:00 than for 22 January and 8 February 2013. The particle formation events on 22 January and 8 February 2013 were detected as starting at 8:30. On 6 February and 7 February 2013, events were detected at around 10:00 (Table 1). The $O_3$ concentrations on the Class I event days showed a steady increase from 7:00, which was followed by concentration fluctuations after 12:00 (Figure 7). The average concentration of $O_3$ on the non-event

days ranged from 12 ppb–22 ppb, increased from 7:00 and was stable after 10:00. The $O_3$ concentration was higher on the Class I event days compared to the non-event days. This observation suggests that photochemical reactions occurring during the campaign could be one of the factors that favours particle formation events. Average CO and $NO_x$ on non-event days ranged from 87 ppb–177 ppb and 4 ppb–19 ppb, respectively.

Isoprene is a volatile organic compound (VOC) that has the potential to contribute to new particle formation and particle growth events. Isoprene is mainly emitted by biogenic activities (i.e., vegetation) [74,75]. In addition to biogenic sources, isoprene is also emitted by anthropogenic sources (i.e., traffic emissions) [76,77]. Biogenic sources of isoprene dominated during the MUMBA campaign [42,45] and will therefore play a stronger role in the formation of secondary organic aerosol. Here, we have used isoprene as a marker of biogenic emission. The concentration of isoprene was high before the particle formation and growth event, but slowly decreased during the event (Figure 7). This is consistent with isoprene acting as a precursor for particle formation. The time series of CO, $NO_x$, $O_3$ and isoprene on the selected Class II event day (9 February 2013) were very similar to Class I event days (Figure 7).

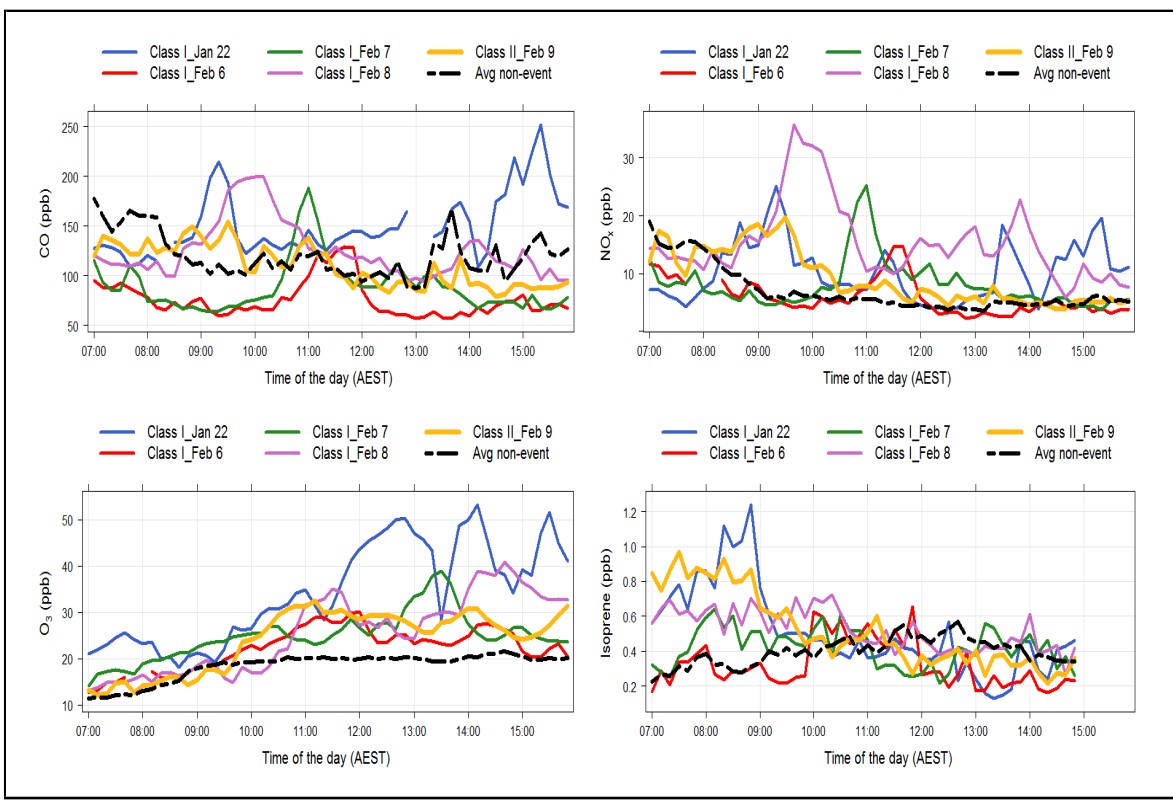

**Figure 7.** Time series of CO, $NO_x$ $O_3$ and isoprene on Class I event days (22 January, 6 February, 7 February and 8 February 2013), a selected Class II event day (9 February 2013) and on the non-event days. Ten-minute averaged data were used. Only measurements from 7:00–15:00 AEST (+10 UTC) are plotted. The proton transfer reaction mass spectrometry instrument used to measure isoprene was calibrated at 15:00.

Results obtained from the comparisons of the available meteorological variables and trace gases on the selected Class II event day and the Class I event days show that conditions were similar. This means that we would have expected to see a particle growth event of the Class II event day (9 February 2013). None of the available information clearly explains the observation of a Class II rather than a Class I event. The influence of relative humidity and unstable wind direction locally could influence the particulate growth and population, even though the observed differences are small.

### 3.2.5. Photochemical Age of Air Masses

Urban atmospheres often include volatile organic compounds such as the aromatic hydrocarbons such as benzene, toluene and xylenes that are important precursors for the formation of secondary organic aerosol [78,79]. The main sources of benzene, toluene and xylenes are anthropogenic, including industrial activities and traffic emissions [80]. The ratio of the concentration of the aromatic compounds can provide useful insights into the sources of the aromatics and the photochemical age of an air mass [81,82]. The toluene to benzene ratio (T/B) can be used as an indicator of traffic emission [83]. The T/B ratios that are greater than 4.5 indicate industrial-originated emission sources [80], whereas T/B ratio values that are within the range of 1.5–3.0 indicate traffic-originated emission [82,84]. The xylenes to benzene ratio (X/B) can act as an indicator of the photochemical age of air masses [85]. Ratio values of X/B that are less than 3.0 imply aged plumes [80].

Concentrations of benzene for all four days (6, 7, 8 and 9 February) (Figure 8a) revealed a similar trend where there were steady increases in concentration in the morning (9:00) and decreases from noon. The concentrations of benzene on 6 February and 7 February were lower than those on 8 February and 9 February, especially before 10:30. Concentrations of toluene on these four days (Figure 8b) were usually less than 1.0 ppb; however, there were high toluene concentrations observed between 8:00 and 8:30 on 6 February. High concentrations of xylenes were observed twice on 7 February (9:30–10:00 and 12:30–14:00) and once on 9 February (10:30), compared to the xylene concentrations observed on 6 February and 8 February (Figure 8c).

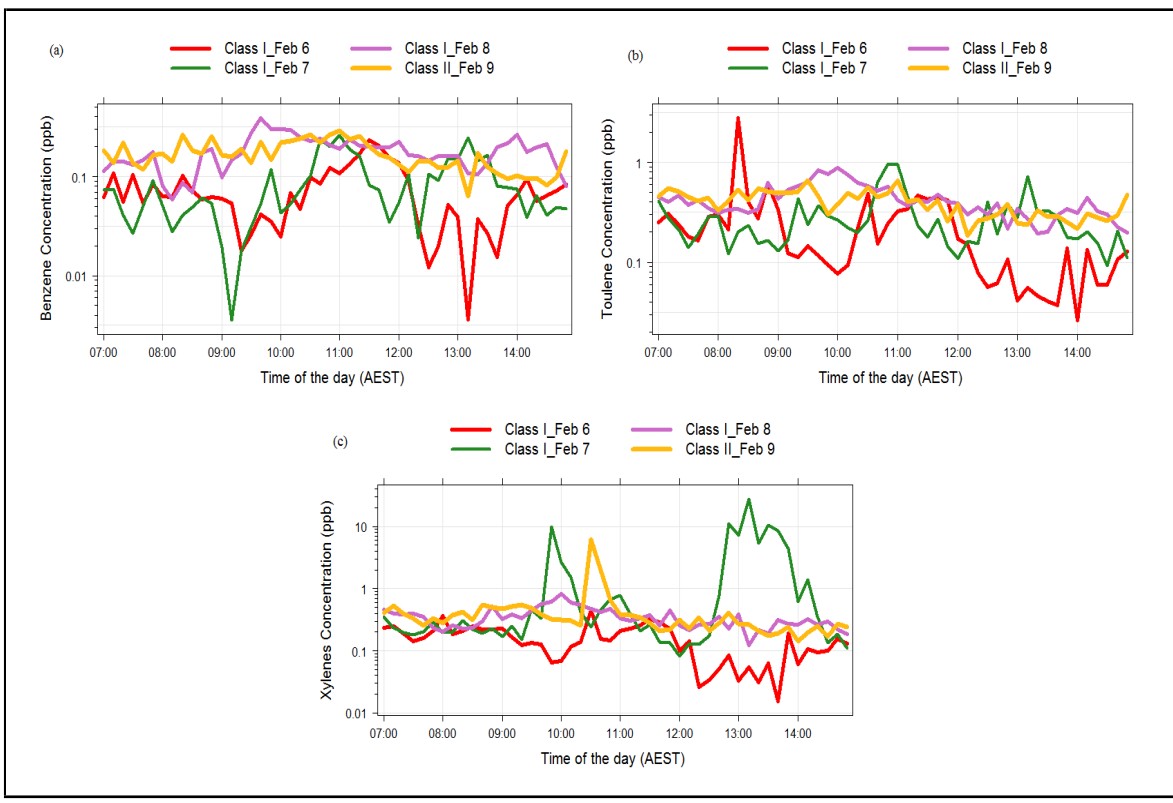

**Figure 8.** Time series of the (**a**) benzene, (**b**) toluene and (**c**) xylenes concentrations on the four consecutive days (6, 7, 8 and 9 February 2013). 6, 7 and 8 February were the Class I event days, and 9 February 2013 was the selected Class II event day. Only measurements from 6:00–15:00 are plotted. Note that y-axes are in log-format. Ten-minute averaged data were used.

Time series of the ratios of the three VOCs are presented in Figure 9. Initially, the xylenes to benzene (X/B) ratios will be discussed to study the influence of the photochemical age of air mass on the Class I particle formation and growth. The X/B ratios on 6 February and 8 February were stable

and consistent throughout the day compared to 7 February and 9 February. There was evidence of fresh plumes (X/B > 3.0) and industrial-originated emissions before the Class I event on 6 February (8:00–10:00). Fresh plumes (X/B > 3.0) and a mixture of industrial and traffic-originated emissions were experienced on 8 February (6:00–8:00). During the relevant time of Class I particle formation and growth on 6 February (10:00 to 13:00) and 8 February (after 8:00 to 14:00), the monitoring station experienced emissions from traffic and aged air masses.

On 7 February, the Class I particle formation and growth event was observed between 10:00 and 14:00 (Figure 2). Before the event (just after 9:00), the monitoring site experienced a mixture of fresh plumes (X/B more than 3.0) and industrial-originated emissions (T/B more than 4.5) (Figure 9). At 10:00, there were high concentrations of smaller particles observed, and the Class I particle formation and growth event were observed (Figure 2). After 10:00, there was a gradual decrease in X/B ratios (less than 3.0), suggesting that the monitoring stations experienced ageing air mass. The Class I particle formation and growth event process continued at this time. After 12:30, the monitoring site experienced additional fresh plumes (X/B ratio reaching more than 100). The Class I particle formation and growth event process was still continuing, but stopped at around 14:00. At this time, the campaign site was still being influenced by traffic-originated emissions at this specific time (Figure 9).

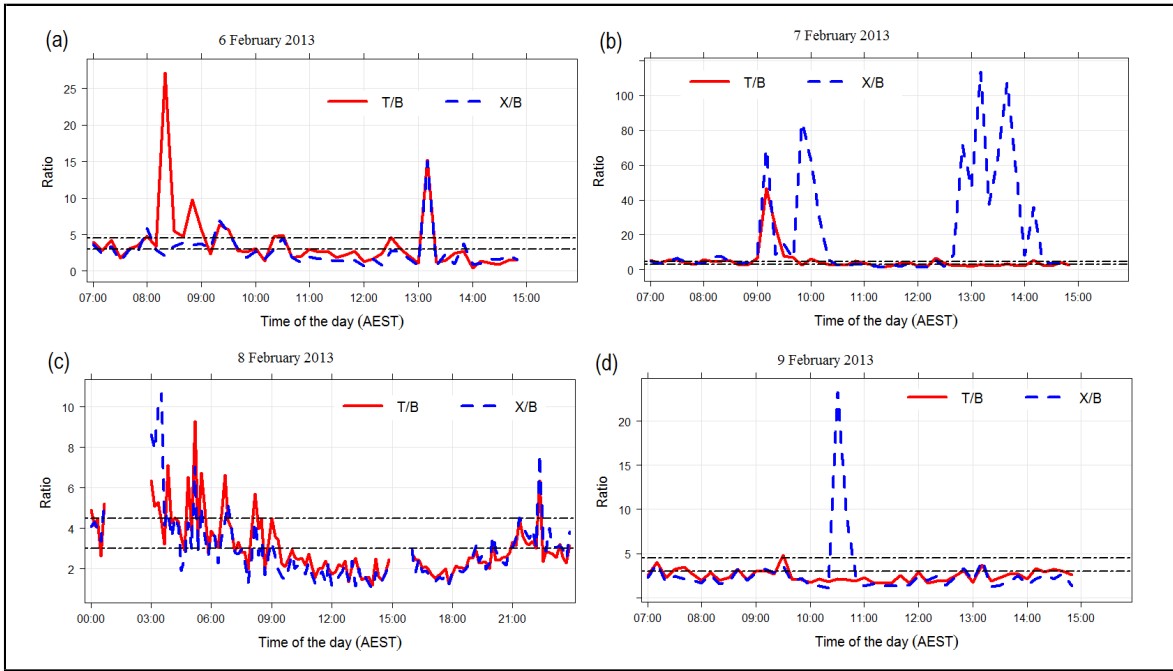

**Figure 9.** Time plots of the ratios of toluene to benzene (T/B) and xylenes to benzene (X/B) on the four consecutive days ((**a**) 6, (**b**) 7, (**c**) 8 and (**d**) 9 February 2013). The black dashed line represents ratio values suggested by Tiwari et al. [80]: Toluene to benzene (T/B) ratios that are greater than 4.5 indicate industrial-originated emission sources [80]. Xylene to benzene (X/B) ratios less than 3.0 imply aged plumes [80]. Note that the y-axes are on different scales. Only measurements from 6:00–15:00 are plotted. Ten-minute averaged data were used.

On 9 February, the Class II particle formation and growth event commenced just after 8:00 (Figure 2). Before the event (just before 8:00), the monitoring site experienced aged air masses (X/B were less than 3.0) (Figure 9) instead of fresh air masses, which were on 7 February. After 8:00, the X/B ratio was less than 3.0 (Figure 9). At the same time, the ratio of T/B was less than 4.5, which suggests that the monitoring station experienced traffic-originated emissions and aged air masses. At around 10:30, the site experienced fresh plumes (X/B greater than 3.0). However, there was still no clear particle formation and growth observed.

Photochemical age of the air mass before the observed particle formation events could be one of the factors that impeded Class I particle formation and growth on 9 February 2013. In this case study, the monitoring site experienced an ageing air mass before the Class II particle formation and growth event was observed. Ageing processes modify the properties of the gaseous precursors for particle formation. Therefore, the properties of aged gaseous precursors may have driven the observation of the particle formation and growth on 9 February 2013.

*3.3. Particle Growth from Smaller Size Particles*

A significant increase in the small particle number concentration over time is a useful marker for new particle formation events. While the smallest diameter measured by the SMPS was 14 nm, we can derive information on smaller particles using data from the condensation particle counter instrument, which measured particles >3 nm. Bursts of small particles (nucleation mode) can either grow (or not grow) into bigger particles. During the aerosol phase of the MUMBA campaign, an event of a burst of small size particles (nucleation mode (<10 nm)) was identified when the ratio between particle number ranging from 3 nm–14 nm ($PNC_{3nm-14nm}$) and particle number ranging from 14 nm–100 nm ($PNC_{14nm-100nm}$) (Equation (9)) was greater than or equal to a threshold value and occurring over 30 min. The threshold used was 1.0, which is the median value of the $PNC_{3nm-14nm}/PNC_{14nm-100nm}$ ratio for the entire dataset. Twelve days of bursts of small particles events were identified over the 22 sampling days.

$$\text{Burst of nucleation mode (<10 nm) size particles} = \frac{PNC_{3nm-14nm}}{PNC_{14nm-100nm}} \geq 1.0 \qquad (9)$$

The growth of small size particles into a detectable size by the SMPS on the Class I particle formation and growth event days can be shown by $PNC_{14nm-660nm}$ and the ratio of $PNC_{3nm-14nm}/PNC_{14nm-100nm}$. A high ratio of $PNC_{3nm-14nm}/PNC_{14nm-100nm}$ indicates a large proportion of small particles (3 nm $< Dp <$ 14 nm) in the particle population.

6 February 2013 was used to illustrate the growth of small size particles into a detectable size (Figure 10). The time series observations of $PNC_{14nm-660nm}$ and $PNC_{3nm-14nm}/PNC_{14nm-100nm}$ on 22 January 2013, 7 February 2013 and on 8 February 2013 are illustrated in the Appendix A (Figure A1). As there were missing $CN_3$ data on 22 January 2013 and on 8 February 2013, the growth of smaller sizes could not be studied on these days.

As illustrated in Figure 10, no peak was observed in the $PNC_{3nm-14nm}/PNC_{14nm-100nm}$ ratio before 7:00. There was also no particle growth observed in the contour plot (Figure 2). This suggests that particle formation and growth processes did not occur for sizes > 3 nm at this time. However, from 8:00, the $PNC_{3nm-14nm}/PNC_{14nm-100nm}$ ratio did increase, followed by a decrease at 10:00 due to a relative decrease in smaller particle concentration and an increase in $PNC_{14nm-660nm}$. Subsequently, a particle growth process was observed in the contour plot after 10:00 (Figure 2). These observations illustrate that particle growth can be observed in the size range 3 nm–14 nm prior to being detectable by the SMPS. The increase in $PNC_{3nm-14nm}/PNC_{14nm-100nm}$ ratio at 13:00 represents a burst of small size particles without an observed growth event. A decrease of the total particles several minutes before a clear particle formation observed is a common observation. This can be due to coagulation [13,71].

*3.4. Effect of Particles Produced During the Class I Event Days on Cloud Condensation Nuclei*

Aerosol particles that activate as cloud droplets in the ambient atmosphere are known as cloud condensation nuclei (CCN). CCN influence the characteristics of clouds by modifying the cloud droplet number concentration, the cloud droplet size, the cloud lifetime, as well as the precipitation processes [9,86–88]. The interaction between aerosol particles and clouds is the source of the largest uncertainties in the quantification of the effect of aerosols on climate [3].

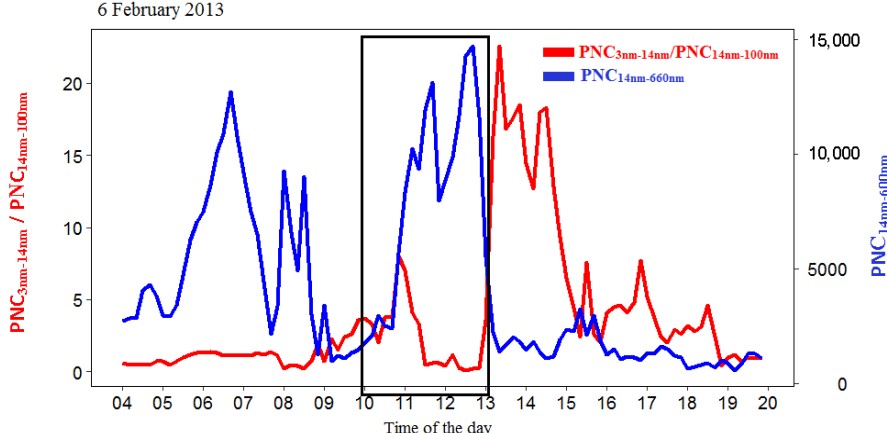

**Figure 10.** Time series of the total particle number between 14 nm and 660 nm ($PNC_{14nm-660nm}$) and the $PNC_{3nm-14nm}/PNC_{14nm-100nm}$ ratio on 6 February 2013. The black rectangular box highlights the growth period. Five-minute averaged data were used in these plots.

Particles less than 50 nm in diameter are inefficient as cloud condensation nuclei [89,90]. Previous studies report that particles with diameters of 50 nm–100 nm have a high potential to be activated and act as cloud condensation nuclei [89,91]. Total CCN increases during particle formation and growth events [92]. A study by Lihavainen et al. [93] in Finland noted that the formation of CCN was related to particle growth and ceased when particle growth stopped.

The CCN concentration and the GMDs of the particles that are equal to and larger than 50 nm observed on the relevant time of particle formation and growth of Class I event days showed a negative correlation, where the strongest correlation was observed on 8 February 2013 (Figure 11). This observation could be caused by a coagulation sink. Kulmala et al. [94] reported that smaller particles (about 10 nm in diameter) coagulate efficiently with larger particles, which therefore increase in particle size. However, the number of smaller particles decreases. The GMDs of particles observed at the relevant time of particle growth on 6 February 2013 were less than 50 nm. Overall, there is insufficient evidence to conclude that particles that are equal to and larger than 50 nm formed during particle formation and growth events in this work were being activated and acted as CCN.

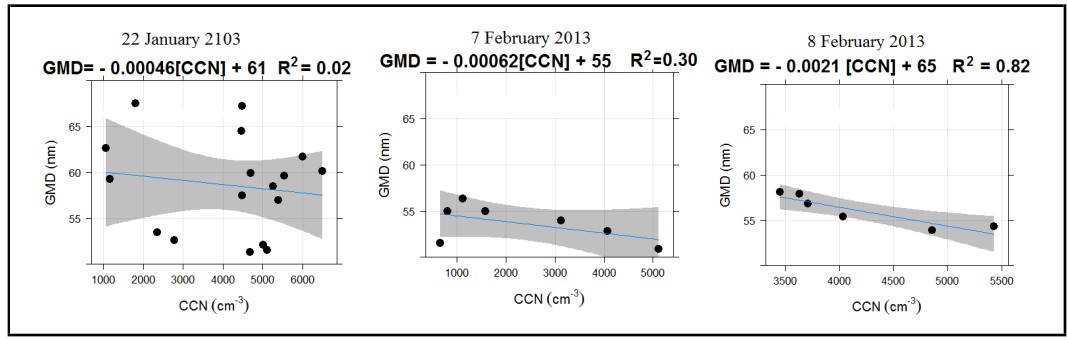

**Figure 11.** Relationship of cloud condensation nuclei (CCN) and geometric mean diameter (GMD) for particles that are equal to and larger than 50 nm observed during the relevant time of Class I particle formation events. The shaded area is the 95% confidence level. 6 February 2013 was not included due to the fact that the observed GMDs on this day were less than 50 nm. Ten-minute averaged data were used.

## 4. Summary and Conclusions

The evolution of particle number size distribution in the range of 14 nm–600 nm was analysed to determine the occurrence frequency of particle formation during the one-month period of aerosol measurements of the MUMBA campaign. Eight particle formation and growth events (four Class I and

four Class II) were observed in this study, which is equivalent to 25% of the total observation days. The Class I events took place in the sunny periods, starting after 8:00 and lasting till 14:30 with an average duration of five hours. These events were classified as "weak" particle formation events. This observation is likely to be related to the meteorological conditions, especially the wind direction experienced at the monitoring site. There was no clear indication that particles with a diameter equal to or larger than 50 nm were activated as cloud condensation nuclei. Surface $O_3$ concentrations increased during the events; meanwhile, a high concentration of isoprene, CO and $NO_x$ occurred before the event and decreased during the events. This suggests that the events were influenced by photochemical reactions and traffic emissions. A mix of oceanic and anthropogenic air masses was also one of the factors that influenced these events. Particle formation and growth events were only observed when air masses originated from the north and northeast sectors. These air masses had travelled from the ocean and passed through populated areas including Sydney. Relative humidity, anthropogenic sulfate and the photochemical age of air masses all potentially played a role in the particle formation and growth events, with potentially the condensation sink limiting growth in Class II particle formation and growth.

**Author Contributions:** Conceptualisation, D.D., S.R.W., C.P.-W., M.K., and R.H.; methodology, D.D, S.R.W, C.P.-W., É.-A.G., R.H., and D.K; validation, R.H., É.-A.G., M.K., and S.R.W.; formal analysis, D.D., É.-A.G., S.R.W., and P.S; investigation, D.D., S.R.W., É.-A.G., R.H., M.K., P.S., and C.P.-W.; data curation, D.D., É.-A.G., and R.H; writing, original draft preparation, D.D.; writing, review and editing, all authors; visualisation, D.D. and B.M.; supervision, S.R.W. and C.P.-W.; project administration, C.P.-W.; funding acquisition, S.R.W. and C.P.-W.

**Funding:** This research was funded by Australia's National Environmental Science Program through the Clean Air and Urban Landscapes hub and from the Australian Research Council Discovery Project DP160101598. The authors would like to thank the University of Wollongong for PhD scholarship support.

**Acknowledgments:** The authors acknowledge all the members of the Centre for Atmospheric Chemistry, University of Wollongong, and CSIRO's Marine and Atmospheric Research group that helped in this campaign. We also acknowledge the NOAA Air Resources Laboratory (ARL) for the provision of the HYSPLIT transport and dispersion model and/or Air Resources Laboratory (http://www.arl.noaa.gov/ready.html). The campaign would not have been possible without the assistance of Kids Uni and the Science Centre that provided the location for the instruments.

**Conflicts of Interest:** The authors declare no conflict of interest.

## Appendix A

This Appendix contains additional figures to help illustrate the growth of smaller size particles during the identified Class I event days and the condensation sink on the Class I and Class II event days.

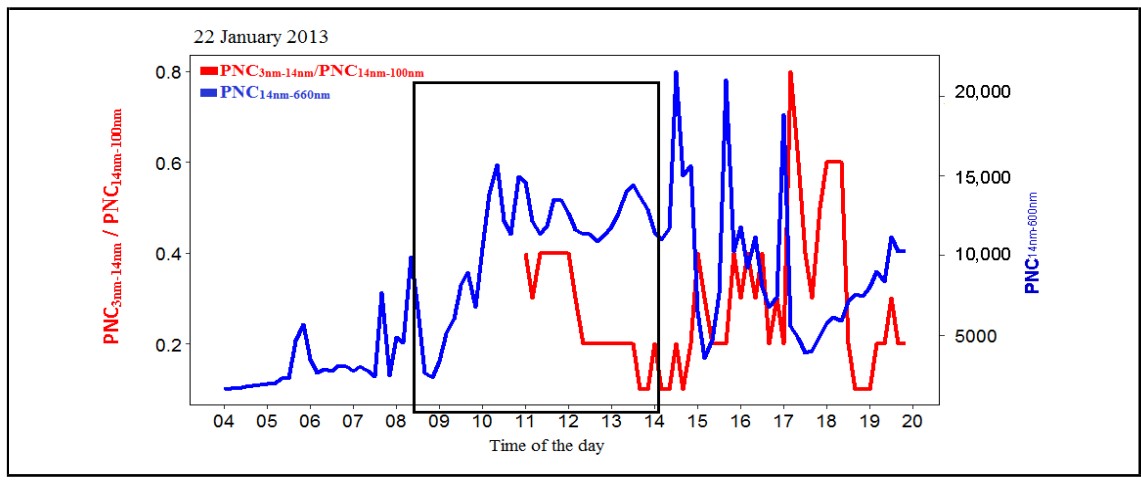

**Figure A1.** Cont.

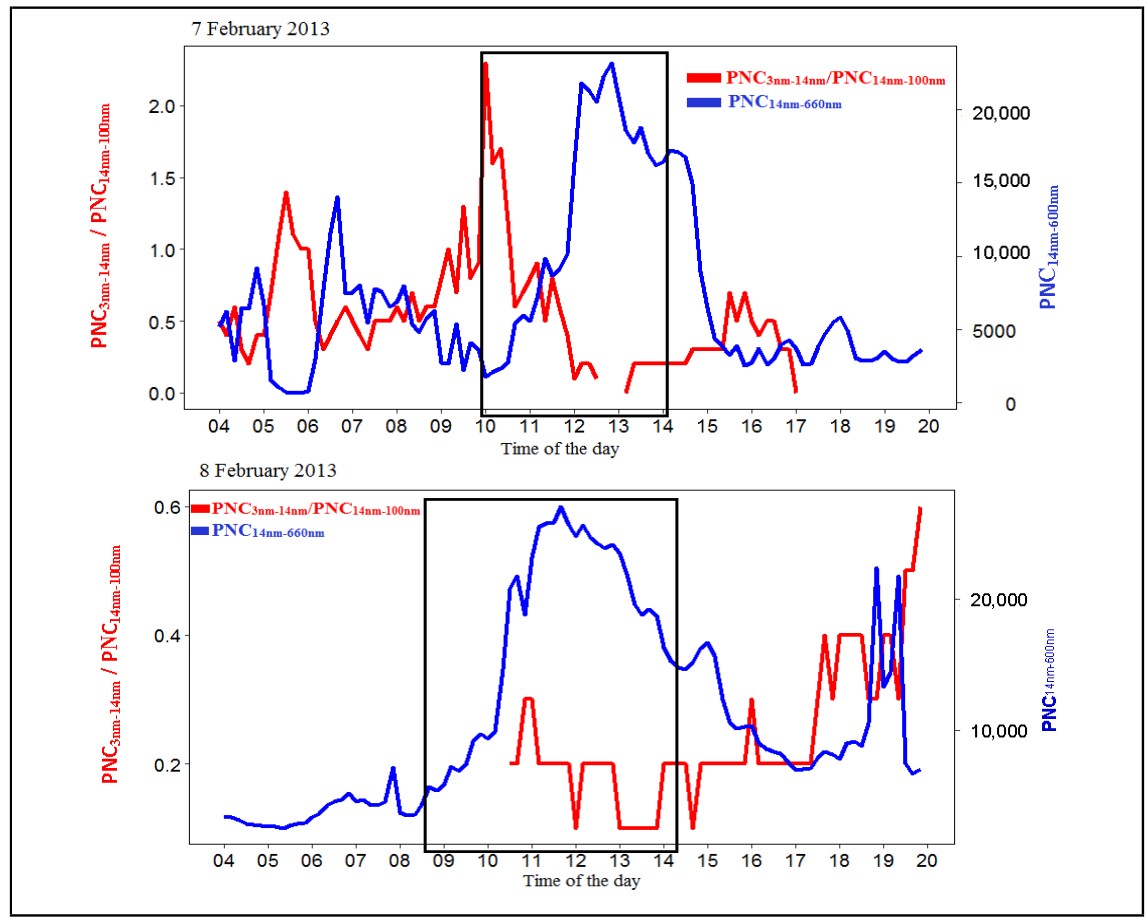

**Figure A1.** Time series of the total particle number between 14 nm and 660 nm (PNC$_{14nm-660nm}$) and the PNC$_{3nm-14nm}$/PNC$_{14nm-100nm}$ ratio on the other three event days (22 January, 7 February 2013 and 8 February 2013). The black rectangular box highlight the growth period. Five-minute averaged data were used in these plots.

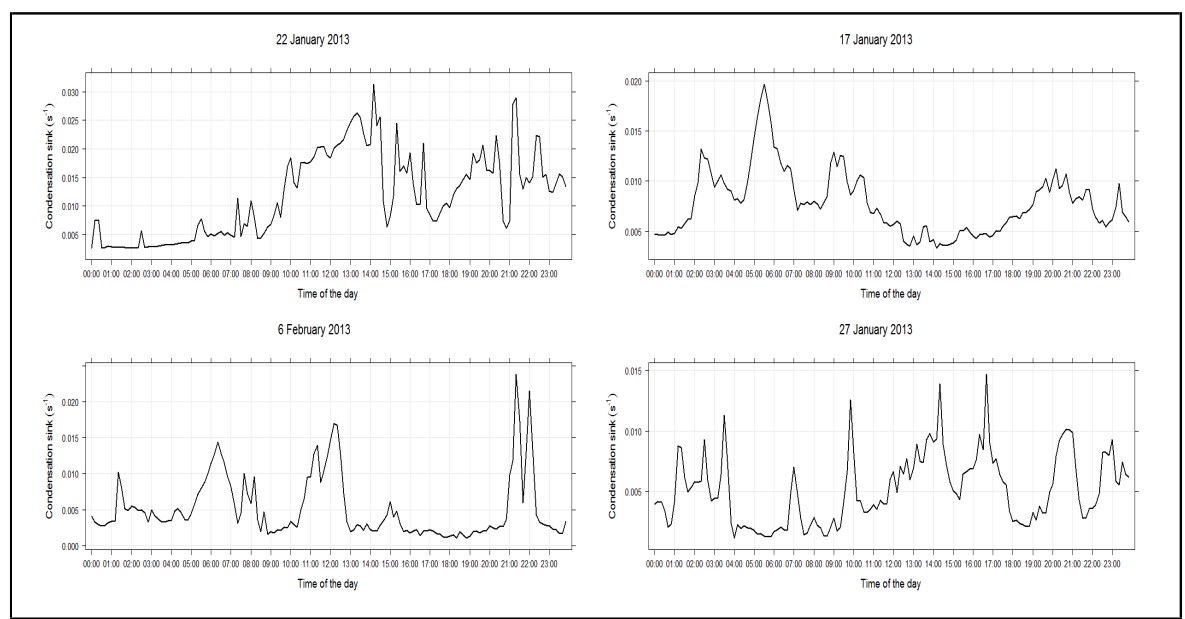

**Figure A2.** Cont.

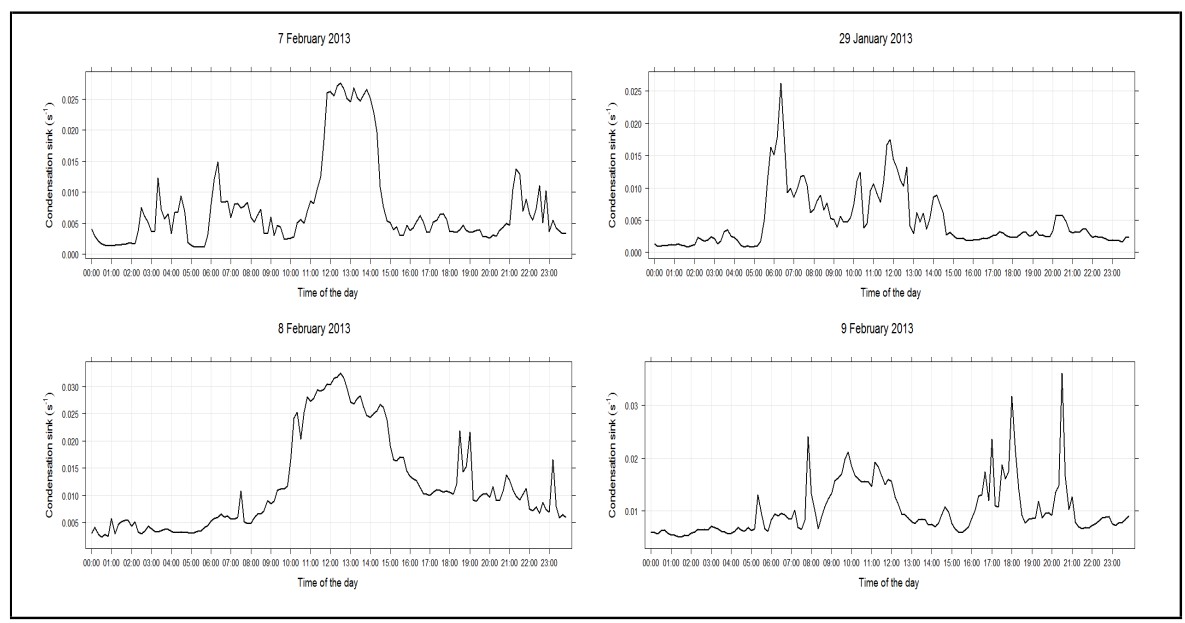

**Figure A2.** Time series of the condensation sink on the Class I (22 January, 6 February 2013, 7 February 2013 and 8 February 2013) and Class II event days (17 January, 27 January 2013 and 29 January 2013 and 9 February 2013).

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
