# Peer review of "Particle Formation in a Complex Environment"

_atmosphere, doi:10.3390/atmos10050275_

Reviewer 1 Report

This is a re-review of manuscript (atmosphere-465308), which presents results from a field campaign in Wollongong, Australia in January-February 2013. The authors have adequately addressed all comments from Reviewer No. 2. This manuscript is now publishable.

Author Response

Thank you reviewer no.1.

There was no comment from reviewer No.1.

However, there are additional corrections added to the manuscript as:

1) "A Class II event is where particle growth is observed, but due to fluctuations in diameter, the growth and formation rate is difficulty to quantify [54]. Page 5, Line 158-159" has been revised to: 

"A Class II event is where particle growth is observed, but due to fluctuations in diameter with time, the growth and formation rate is difficult to uniquely quantify [54]." (Page 2, line 162, Line 164)

 2)  "(SO24-(ss))/ (g MSA)"has been added to  Page 8, Line 208

Together with this email, I have attached the Revised manuscript in PDF.

Reviewer 2 Report

Very nice job addressing my most major comments.

Only a very few minor points remain.

In the literature on NPF in Australia, what did the newly referenced studies find?

How was the 10% uncertainty in anthropogenic sulfate fraction arrived at?

Lines 276-280 now repeat (in less detail) lines 264-271.

line 374 "would" is a bit strong, perhaps "may"

Author Response

Thank you reviewer No.2

Below are the responses for the comments from reviewer No. 2:

Comment 1: The newly referenced studies on NPF in Australia is by Pushpawela et al.(2018).

Response: A recent study on NPF events in the urban environment in north-eastern Australia by Pushpawela et al. [34] reported that 41% of the days contained NPF events. These were identified in one-year of measurements using a neutral cluster and air ion spectrometer (NAIS). The highest occurrence of NPF events was during summer, with a starting time between 8:00 am and 8:30 a.m. Other previous NPF studies in the urban environment in Australia include Salimi et al. [35], Cheung et al. [36] and Mejia et al. [37]. (Page 2, Line 42-Line 47)

Comment 2:  How was the 10% uncertainty in anthropogenic sulfate fraction arrived at?

Response: Clarification has been added to the manuscript:

"20% from sea salt and 10% from biogenic sources. The composition of  sea salt is well established, and even a 50% error in the biogenic contribution would only change the anthropogenic estimate by 5%.We therefore ascribe a conservative estimate of the uncertainty in the anthropogenic fraction of 10%." (Page 9,Line 215-Line 218)

Comment 3:  Lines 276-280 now repeat (in less detail) lines 264-271.

Response: Line 276-280 has been revised to :

"The temperature and wind speed are very similar for all days. The global irradiance on the selected Class II event day was similar to that observed on the 22nd January, 2013." (Page 11, Line 281-Line283)

Comment 4: line 374 "would" is a bit strong, perhaps "may"

Response: "would" has been changed to "may" (Page 16, Line 377)

Additional corrections

Additional corrections added to the manuscript as:

1) "A Class II event is where particle growth is observed, but due to fluctuations in diameter, the growth and formation rate is difficulty to quantify [54]. Page 5, Line 158-159" has been revised to: 

"A Class II event is where particle growth is observed, but due to fluctuations in diameter with time, the growth and formation rate is difficult to uniquely quantify [54]." (Page 2, line 162, Line 164)

2)  "(SO24-(ss))/ (g MSA)"has been added to  Page 8, Line 208

Together with this email, I have attached the Revised manuscript in PDF.

This manuscript is a resubmission of an earlier submission. The following is a list of the peer review reports and author responses from that submission.

Round  1

Reviewer 2 Report

    This manuscript (atmosphere-465308) presents results from a field campaign in Wollongong, Australia in January-February 2013. A Scanning Mobility Particle Sizer (SMPS), Condensation Particle Counter (CPC), and Cloud Condensation Nucleus (CCN) counter were deployed to measure particle size distributions, number concentrations, and cloud droplet activation. The authors observed nanoparticle events of some sort on 25% of measurement days. These event days are subdivided into weak new particle formation event days (Class I) and nanoparticle event days lacking growth (Class II). The authors then explore potential sources of new particles by correlating to meteorological and chemical measurements made at the site.

    The manuscript is within the scope of Atmosphere. However, in its current form, this manuscript cannot be published. As discussed in more detail below, this manuscript lacks any information about certain basic parameters governing new particle formation and instead tries to explain particle formation through correlations with likely minor contributors. The manuscript may be publishable once all the below major comments are adequately addressed. However, addressing these comments will ultimately result in a new manuscript.

Comments:

1.    While most of the comments below relate to major issues with the manuscript, by far the most significant is that the manuscript fails to discuss the role of condensational sink. For a strong new particle formation day to occur, typically three conditions must be met: 1) high solar irradiance to photochemically produce condensable species, 2) low condensational sink to ensure new particles are formed rather than condensation of low-volatility compounds onto existing particles, and 3) a threshold sulfuric acid concentration, as sulfuric acid is a key player in new particle formation, especially in impacted environments like the one studied here. This manuscript discusses to some extent points 1) and 3) but completely ignores discussion of point 2), the condensational sink. Any possible revision must include calculations and discussion of condensational sink. This discussion could also enhance the information gained from Class I vs. Class II days (i.e. do Class II days have higher condensational sink, so the events or particle growth are weaker?) as well as the discussion of sulfate (overall, sulfate concentrations are higher on event days, but there are many non-event days when sulfate seems to be high as well – are these days with higher condensational sink?)

2.    Instrumentation: In its current form, the description of instrumentation used is insufficient. Some (non-exhaustive) examples of additional information that should be contained in the manuscript include: the brand/model number of the SMPS, the condensing vapor used in the condensation particle counter, and the time and size resolution of the SMPS. The authors refer to previous work, but information along these lines must be included in the current manuscript.

3.    Organization of the manuscript should be improved. As an example, the authors discuss Class I and Class II particle formation events. However, only Class I events are defined in the experimental section. Class II events are defined only on line 126 (the Results section). Moreover, the manuscript would benefit from a restructure where the most important results are presented first. The authors will improve clarity and coherency in the manuscript by placing (and expanding) the discussion on sulfate closer to the beginning of the results section. As no obvious correlations are observed for most of the other parameters (e.g. CO, NOx) the authors should de-emphasize these aspects.

4.    The Class II event days look more like plume events rather than new particle formation events, as they show a burst of nanoparticles for some period of time with minimal growth. Do the back trajectories inform about where the air masses passed over (e.g. power plants) that might provide insight into the sources of these particles?

5.    The discussion of Figure 3 is confusing. The small/large particle ratio in fact decreases at the start of the new particle formation event and only increases much later when the event is nearly over. Comparing this plot to the SMPS plot in Figure 2, it looks like the sub-14 nm particles remain present but there is a substantial decrease in the number of larger particles that causes this ratio to change. Therefore, it is not clear that this ratio is reporting what the authors state should be happening.

6.    The entire discussion on the CCN measurements does not seem to provide any clear result. What should a reader take from this discussion? Why is geometric mean diameter inversely correlated with CCN number concentration?

7.    For the plots of meterological data, it is hard to see any clear trend other than the increase in solar irradiance on event days. This may be clarified somewhat by showing the average and range for non-event days compared to the average and range of event days. Showing only an average for the non-event days (which presumably also includes the Class II days) obscures the likely substantial variability in conditions on non-event days.

8.    The discussion of CO, NOx, O3, and Isoprene does not seem to add much to the manuscript as no clear trends are observed. All data ultimately cluster around the average. This would probably be event more the case if a range were given for the averages on non-event days.

9.    The title of the paper does not seem to represent the content: the mechanisms of particle formation are not identified. Only some correlations are explored.